# Early observations from the HIV self-testing program among key populations and sexual partners of pregnant mothers in Kampala, Uganda: A cross sectional study

Esther Nasuuna[1]*, Florence Namimbi[1], Patience A. Muwanguzi[2], Donna Kabatesi[3], Madina Apolot[3], Alex Muganzi[1], Joanita Kigozi[1]

**1** Infectious Diseases Institute, College of Health Sciences, Makerere University, Kampala, Uganda,
**2** Department of Nursing, School of Health Sciences, College of Health Sciences, Makerere University, Kampala, Uganda, **3** Division of Global HIV and TB, Centres for Disease Control and Prevention, Kampala, Uganda

* enasuuna@gmail.com

## Abstract

### Background

HIV self-testing (HIVST) was adopted for key populations (KPs) and sexual partners of pregnant and lactating women (mothers) in Uganda in October 2018. We report early observations during HIVST implementation in Kampala, Uganda.

### Methods

HIVST was rolled out to reach those with unknown HIV status at 38 public health facilities, using peer-to-peer community-based distribution for female sex workers (FSW) and men who have sex with men (MSM) and secondary distribution for mothers, who gave HIVST kits to their partners. Self-testers were asked to report results within 2 days; those who did not report received a follow-up phone call from a trained health worker. Those with HIV-positive results were offered confirmatory testing at the facility using the standard HIV-testing algorithm. Data on kits distributed, testing yield, and linkage to care were analysed.

### Results

We distributed 9,378 HIVST kits. Mothers received 5,212 (56%) for their sexual partners while KPs received 4,166 (44%) (MSM, 2192 [53%]; FSW1, 974, [47%]). Of all kits distributed, 252 (3%) individuals had HIV-positive results; 126 (6.5%) FSW, 52 (2.3%) MSM and 74 (1.4%) partners of mothers. Out of 252 individuals who had HIV-positive results, 170 (67%) were confirmed HIV-positive; 36 (2%) were partners of mothers, 99 (58%) were FSW, and 35 (21%) were MSM. Linkage to treatment (126) was 74%.

**Data Availability Statement:** The datasets analysed to inform this study have been provided as S1 Dataset.

**Funding:** The authors received no specific funding for this work.

**Competing interests:** The authors have declared that no competing interests exist.

**Abbreviations:** AIDS, Acquired Immune Deficiency Syndrome; ART, Ant-Retroviral Therapy; CDC, Centres for Disease Control; FSW, Female sex worker; HIV, Human Immunodeficiency Virus; HIVST, HIV Self testing; HTS, HIV Testing Services; IDI, Infectious Diseases Institute; IPV, Intimate Partner Violence; KP, Key populations; MCH, Maternal Child Health; MSM, Men who have sex with Men; PLHIV, People Living with HIV; WHO, World Health Organisation.

## Conclusions

HIVST efficiently reached, tested, identified and modestly linked to care HIV positive FSW, MSM, and partners of mothers. However, further barriers to confirmatory testing and linkage to care for HIV-positive self-testers remain unexplored.

## Introduction

Globally, approximately 21% of people living with the Human Immunodeficiency Virus (HIV) (PLHIV) are unaware of their HIV-positive status [1]. According to the Joint United Nations Programme on HIV/AIDS (UNAIDS) country report of 2019, 89% of PLHIV in Uganda were aware of their HIV status [2]. The first 95 in the UNAIDS 95-95-95 targets for HIV epidemic control by 2030 states that 95% of all PLHIV should know their HIV status [3]. To increase access to antiretroviral therapy (ART), hence achieve HIV epidemic control, PLHIV need to know their status first [4]. Key populations (KPs), who include men who have sex with men (MSM), sex workers, transgender people and injecting drug users, account for 28% of all new HIV infections in Eastern and Southern Africa [1]. Despite KPs being disproportionately affected by HIV, they continue to lag behind other populations in the first 95 in Eastern and Southern Africa with only 54% of MSM living with HIV knowing their HIV status [1,5]. This has led to delay in epidemic control since identification is the first step [6]. Partners of pregnant and lactating mothers (mothers) are an underserved group who also need to know their HIV status [7].

HIV self-testing (HIVST) allows individuals to use a rapid test to determine their HIV status in a private setting [8]. Literature shows that HIVST is a feasible intervention for reaching KPs and partners of pregnant and lactating mothers who are underserved by other testing approaches [9–11]. It can ensure better uptake, encourages earlier diagnosis, and potentially increases access to HIV testing services (HTS) [11]. A trial in central Uganda to reach partners of pregnant and lactating women found that HIVST reached more partners than conventional facility-based testing [12]. In Kenya, a similar trial had acceptance rates of 91% among sexual partners of pregnant mothers [13] and 94% among the general population [4]. HIVST also increased uptake for HIV testing among MSM and their partners in South Africa, they found the strategy convenient, private, and empowering, preferable to clinic-based testing [14]. A study in Uganda also showed high acceptability of 82% by fishermen [15] and MSM [16]. Other studies have also shown that HIVST can increase uptake in couples' testing for HIV [11,13,14]; in addition to promoting knowledge of partner status, HIVST helps to inform decision making around sex [13,17].

Uganda adopted HIVST as a testing approach for KPs and mothers' partners in 2018 [18]. This policy adoption change was informed by studies that showed high acceptability and demonstrated validity of results, which were similar to supervised tests [15,19]. These were randomized controlled trials that informed the scale up in public health facilities that we report on. The oral fluid OraQuick (OraSure Technologies, Inc. Bethlehem, PA) HIVST kits were recommended for use. OraQuick has been found to have a high sensitivity 100% and specificity (100%) in a Ugandan study compared to the blood based tests [20]. HIVST was first scaled up for partners of pregnant and lactating mothers and KPs at selected high-volume public and private not for profit health facilities in September 2018. HIV self-testing kits were distributed in the facility and community for use by partners of pregnant and lactating mothers and KPs respectively. We report the early observations from implementation of this program among KPs and partners of mothers at the public sites in Kampala, Uganda.

## Methods

### Study design

This was a cross sectional study of clients who participated in the HIVST program in the Infectious Disease Institute (IDI) supported Kampala region from October 2018 to June 2019.

### Study setting

The study was conducted at all 38 urban public health facilities in Kampala that were part of the HIVST scale-up in October 2018—June 2019. Participants were individuals aged 18 years and above as well as emancipated minors (individuals aged16-<18 years who have children or are heads of households) as per national implementation guidance.

### Program description

HIVST was implemented using peer-to-peer model for KPs and secondary distribution for partners of consenting pregnant and lactating mothers. The KP peers are members of KP community who were recruited to support with demand creation for and distribution of HIVST kits. The pregnant and lactating mothers delivered HIVST kits to their partners with unknown HIV status at home. Distribution to pregnant and lactating mothers was done by midwives. Selected health workers, including midwives, and peer distributors at participating health facilities received 3-day trainings prior to distribution of HIVST kits. The training focused on self-test kit use, importance of HIV positive result confirmation, management of tests including storage and documentation. Mothers were tested at the health facilities using the national testing algorithm as recommended by the Ugandan Ministry of Health. Consenting mothers, only those with partners of unknown HIV status were offered HIVST. Education on how to use HIVST kits was provided in the local language, to mothers prior to distribution. Education was done through health talks, practical demonstrations, and videos. All mothers were screened for intimate partner violence (IPV) before HIVST kits were given, and those at risk were supported accordingly by a counsellor trained in handling IPV. Those who were in danger of IPV were either supported by health worker to prepare the partner to receive the test or left out if the mother felt this intervention was not enough to protect her from IPV. The telephone numbers for both mothers and the partners to whom the HIVST kits were to be distributed for use were recorded for follow-up.

KP peers trained in HIVST at the facility, distributed HIVST kits in the community to KPs with self-reported unknown HIV status. The KPs were screened with the national HTS screening tool for adults that assesses risk as well as duration since the last test. The KPs reached were FSW and MSM who lived in the same communities as the peers and were known to them. Peers received HIVST kits for distribution from designated health facilities. Distribution was mostly done at KP hotspots such as bars and KP dens, which are convenient meeting points for KPs. Telephone numbers for both peers and KPs that received HIVST kits for use were recorded for follow-up purposes.

All HIVST kits distributed in the facility and community were recorded in the national HIV Self-test Kit distribution logs. In the community, the logs were accessible only to peers. Two days after receiving HIVST kits, mothers and peers were asked to confirm delivery of the kits to recipients through a phone call. Recipients were individuals (MSM, FSW and partners of mothers) who received kits from either peers or mothers for use by themselves to determine their HIV status. Upon confirmation of delivery of kits, recipients were called within two days and asked to report their results, through a phone call to a designated health worker. Recipients who did not report results within two days, were followed-up through a phone call by a

trained health worker to determine kit utilisation and to obtain the results. Recipients who reported HIV-positive results were offered confirmatory testing using the national HIV testing algorithm at the nearest health facility of their choice. They were required to present OraQuick paddle to the health facility as proof of the positive HIVST result. Those who were unable to present at the facility for confirmatory testing within seven days were followed up and offered confirmatory testing through a community visit.

For this study, a descriptive analysis was conducted using Microsoft Excel (Redmond, WA). Quantitative data on number of kits distributed from October 2018 to June 2019, target population, testing yield, and linkage to care are presented as percentages. Linkage to care was defined as receipt of ART, which happened either in the facility where HIVST kits were distributed or other facilities as long as the recipient provided an ART number as proof.

### Ethics approval and consent to participate

Ethical approval was obtained from the School of Public Health Ethics Committee at Makerere University College of Health Sciences (reference number 710), Uganda National Council of Science and Technology (UNCST No: HS553ES) and CDC's Centre for Global Health (ref. number: 2019–175). As a retrospective analysis of de-identified routine data, a waiver of consent was obtained.

## Results

Staff at participating facilities distributed 9,378 HIVST kits. Most (6965 [74%]) to women aged 16 to 24 years (4496 [48%]). Kits used by male sexual partners of mothers were 5212 (56%). In the community, key populations received 4166 (44%) kits: MSM 2192 (23%) FSW 1974 (21%). See Fig 1.

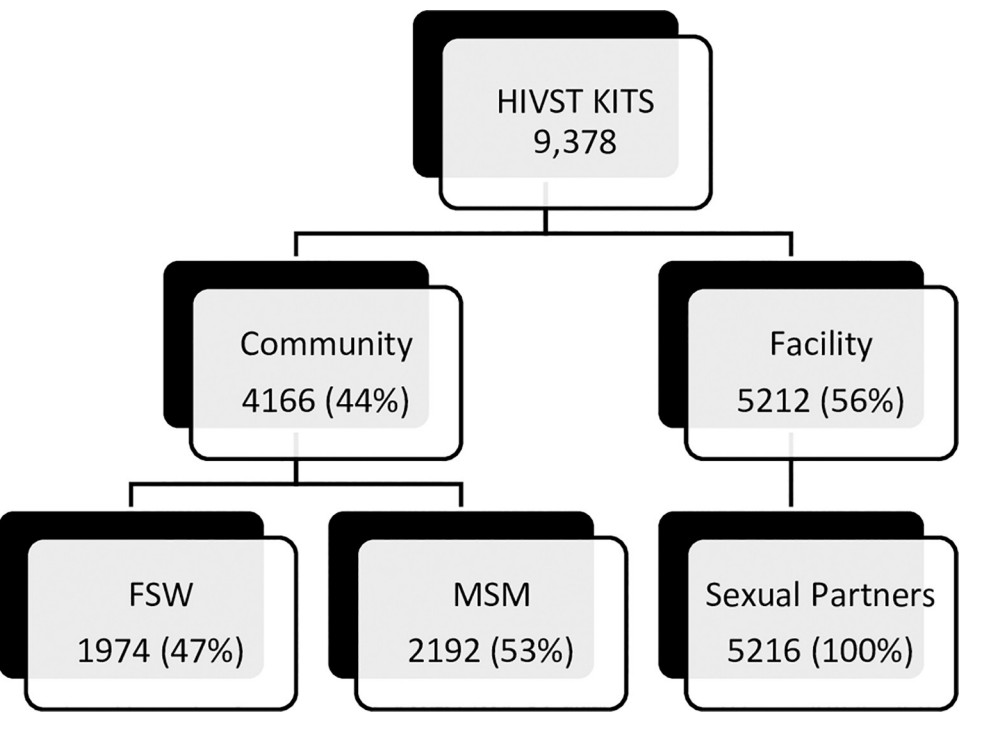

**Fig 1. Flow chart showing the client distribution.**

**Table 1. Characteristics of the distributors and recipients of HIV self-test kits (9378) in Kampala, Uganda (October 2018–June 2019).**

| | Number of HIV kits N (%) |
|---|---|
| **Distribution approach** | |
| **Facility (For partners)** | 5212 (56) |
| **Community (KPs)** | 4166 (44) |
| **Sex** | |
| **Male** | 2413 (26) |
| **Female** | 6965 (74) |
| **Age in years of distributors** | |
| **16–24** | 4496 (48) |
| **25–34** | 4022 (43) |
| **35–44** | 745 (8) |
| **≥45** | 115 (1) |
| **Recipients** | |
| **Mother's partner** | 5212 (56) |
| **FSW** | 1974 (21) |
| **MSM** | 2192 (23) |
| **HIVST Result** | |
| **Negative** | 9126 (97) |
| **Positive** | 252 (3) |

Abbreviations: FSW, female sex workers; MSM, men who have sex with men.

Of the 9,378 HIVST kits distributed to partners and KPs, 9,126 (97%) recipients reported HIV-negative results, and 252 (3%) 95% confidence interval (2.3 to 3.03%) reported HIV-positive results. See Table 1.

The highest yields were among the KPs 126 (6.5%) FSW, and 52 (2.3%) MSM and lowest among the partners of mothers 74 (1.4%) Among participants who reported their HIV status, 17 (7%) already knew they were HIV positive but had not disclosed to their partners or the peer distributing the kits.

Of the 170 (67%) recipients who underwent confirmatory HIV testing, 36 (21%) were partners of mothers, 99 (58%) were FSW, and 35 (21%) were MSM; only 126 (74%) were linked to care and initiated treatment. This percentage drops to 50% (126/252) if we assume that all who reported an HIV positive result were truly HIV positive. The highest number of participants linked to care was FSW at 78 (79%), followed by MSM 26 (74%) and sexual partners 22 (61%). See Fig 2.

## Discussion

In this study we report early observations from implementing the HIVST program at 38 sites in Kampala, Uganda. We found that facility-based distribution was more utilized and HIVST reached more partners of mothers than it did KPs. The testing yield was highest among FSW at 6.5% and linkage to care was suboptimal across all the populations.

We found that most of the kits were distributed by mothers to their sexual partners. This is similar to a study in Kenya where mothers in the antenatal and postpartum clinics distributed 59% of kits to their sexual partners [13]. In Malawi, delivery of HIVST kits by pregnant women to their partners was also acceptable to both the women and their partners [7]. This shows that it is a feasible option to increase HIV testing among partners of mothers.

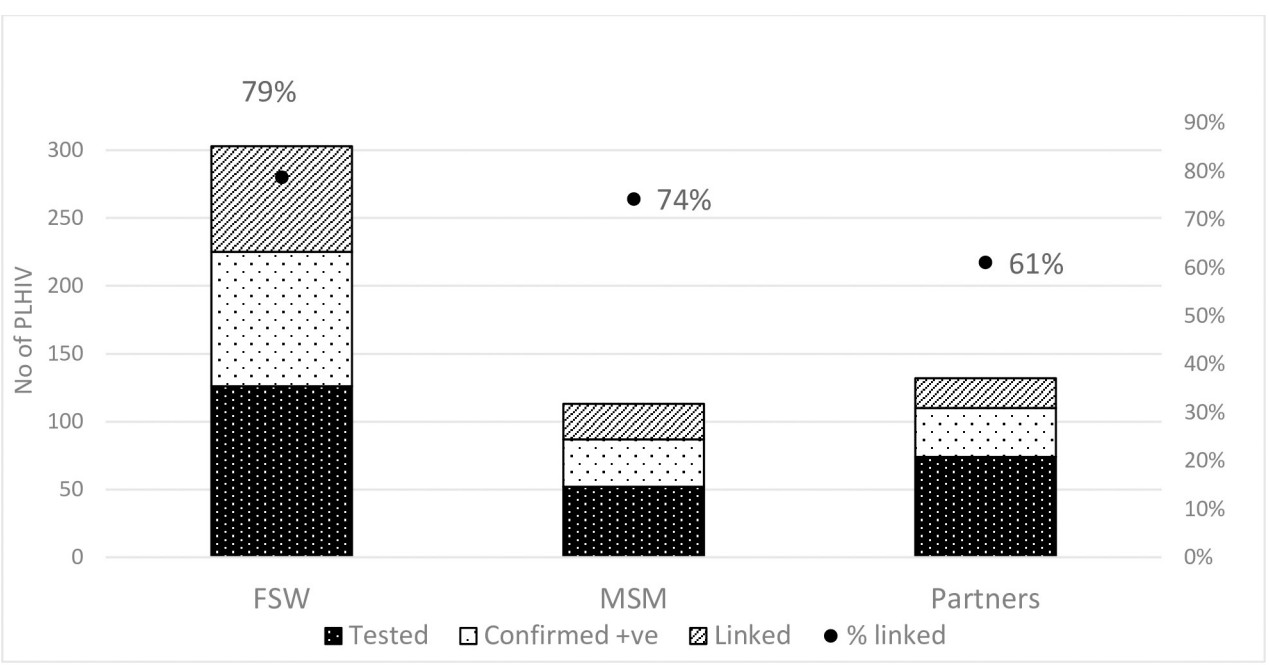

Abbreviations: FSW, female sex workers; MSM, men who have sex with men.

**Fig 2. Linkage to care of recipients who tested HIV positive through use of HIV self-test kits (HIVST) in Kampala, Uganda (October 2018–June 2019).**

HIVST kits were utilized by the FSW and MSM. Previous studies have shown that these groups find peer distribution of HIVST kits acceptable and it also increases their chances of testing for HIV [21–23]. A cluster randomised trial in Uganda found that FSW who received HIVST kits from their peers were more likely to test at the recommended intervals [21]. Programs should consider peer- based distribution of HIVST to increase testing among MSM and FSW.

The study also found that only 3% of recipients of HIVST kits reported an HIV positive result. Our findings are similar to the Kenyan study among male partners of mothers, which also reported a 3% positivity rate [17]. This positivity rate was much lower than reported in a similar study in Blantyre, Malawi, which showed a prevalence of 11.8% in the community [24]. However, this could be due to the fact that Malawi with an HIV prevalence of 10.6% has a higher prevalence than Uganda of 6.2% [25,26]. The prevalence among the MSM was 2.3% in this study. This is much lower than the one in South Africa that found 5.5% positivity among MSM [14]. FSW had the highest positivity rate at 6.5%. A small percentage of 7% among all who received HIVST kits knew about their HIV positive status but went ahead to take a HIVST kit and to retest. This proportion was far below the one reported in Malawi of 26% [24].

The linkage to care was far below the program target of 95% but higher than the rates reported in studies in Malawi (56%) [24] and South Africa (67%) [14]. However, it would be lower than Malawi if we considered 50% of all who tested positive on HIVST as done in the Malawi study. The rate in South Africa was among only the MSMs that participated in the study. A systematic review and meta-analysis among randomised controlled trials showed that linkage to care and confirmatory testing even in the highly controlled trials were problematic

[27]. A pilot study in Uganda showed 100% linkage to care for MSM when followed up by peers [23]. A systematic review in sub-Saharan Africa also showed higher linkage to care rates when active referral was employed [28]. In the Uganda HIVST program, strategies used to improve linkage to care included referral to facility, phone call reminders, community ART initiation, and home visits for linkage. A study in Malawi showed that home initiation of ART after HIVST improved linkage to care [29] and could help reduce missed opportunities for linkage to care in Uganda. Optimizing these strategies may improve future linkage rates.

Our data suggest that increasing the number of HIVST recipients who undergo confirmatory testing, and are linked to care would increase number of PLHIV receiving ART. These findings are only applicable to settings similar to Kampala, Uganda.

## Limitations of the study

The secondary data used in this analysis was program data, which was not collected for research purposes. As such data that could have been reported may have been missing and not analysed. We would have liked to know the people that refused to join the program and the reasons why people dropped off at every stage of the program. We used self-report to record HIV negative results; proof of test result was not required as per the national guidelines. Additionally, we relied on mothers or the KP peers to determine the HIV status of the partner or KP as unknown. It is possible that the HIV status was known to the partner or KP and they just hadn't disclosed as seen by the 17 HIV positive who reported already being in care.

## Conclusion

HIVST is a promising intervention that could be used to identify individuals with previously undiagnosed HIV among KPs and sexual partners of mothers who are hard-to-reach with conventional testing. However, confirmatory HIV testing and linkage to care for those who report a positive HIVST result remains a challenge particularly among non-KPs. Scaling up community confirmatory HIV testing and ART initiation could increase linkage to ART. A cost effectiveness analysis should be done to guide adoption of this recommendation. Further exploration of self-testers' perceptions and attitudes to confirmatory testing and linkage to care and treatment could inform future strategies and national policy changes to improve linkage to care.

## Supporting information

**S1 Checklist. Strobe checklist for cross sectional studies.**
(DOC)

**S1 Dataset. Data set containing the data that was used for this analysis.**
(XLSX)

## Acknowledgments

The authors acknowledge the health workers and peers at the implementing facilities where this data was collected. This program has been supported by the President's Emergency Plan for AIDS Relief (PEPFAR) through CDC, Uganda under the terms of GH001294. The findings and conclusions in this report are those of the authors and do not necessarily represent the official position of the Centres for Disease Control and Prevention.

## Author Contributions

**Conceptualization:** Esther Nasuuna, Florence Namimbi, Patience A. Muwanguzi, Joanita Kigozi.

**Formal analysis:** Esther Nasuuna.

**Resources:** Alex Muganzi.

**Supervision:** Alex Muganzi, Joanita Kigozi.

**Writing – original draft:** Esther Nasuuna.

**Writing – review & editing:** Esther Nasuuna, Florence Namimbi, Patience A. Muwanguzi, Donna Kabatesi, Madina Apolot, Joanita Kigozi.

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
