## [Decision Letter · Decision Letter 0]

28 Sep 2021

PGPH-D-21-00581

Early Observations from the HIV Self-Testing Program among Key Populations and Sexual Partners of Pregnant Mothers in Kampala, Uganda

Dear Dr. Nasuuna,

Thank you for submitting your manuscript to PLOS Global Public Health. After careful consideration, we feel that it has merit but does not fully meet PLOS Global Public Health’s publication criteria as it currently stands. Therefore, we invite you to submit a revised version of the manuscript that addresses the points raised during the review process.

EDITOR:

This study describes original work by Nasuuna et al on HIV self-testing in Uganda which reports relevant data. A number of methodological flaws are present, which are however possible to address. The main conclusion is, that HIV self test efficiently reached, tested, identified a number of relevant risk groups and to some extent linked them to care, this conclusion is supported by the presented data.

The article does however not adhere to appropriate reporting guidelines such as STROBE and this must be improved, a long with revision of all major comments by reviewers. Minor comments should be adhered to as well but may be discussed if relevant.

We look forward to receiving your revised manuscript.

Kind regards,

Christian Wejse, MD, PhD, Assoc.Prof

Academic Editor

Journal Requirements:

1. Please provide separate figure files in .tif or .eps format only, and remove any figures embedded in your manuscript file. 

2. Please update the completed 'Competing Interests' statement, including any COIs declared by your co-authors. If you have no competing interests to declare, please state "The authors have declared that no competing interests exist". Otherwise please declare all competing interests beginning with the statement "I have read the journal's policy and the authors of this manuscript have the following competing interests:"

3. In the online submission form, you indicated that "Data is available from the corresponding author upon request."

4. Please amend your detailed Financial Disclosure statement. This is published with the article, therefore should be completed in full sentences and contain the exact wording you wish to be published.

Additional Editor Comments (if provided):

STROBE guidelines should be adhered to, eg describe the design in title and abstract. Please complete the STROBE checklist for cross-sectional studies and submit with a revised version along with a flow chart of patients enrolled

Reviewers' comments:

Reviewer's Responses to Questions

**Comments to the Author**

1. Does this manuscript meet PLOS Global Public Health’s publication criteria? Is the manuscript technically sound, and do the data support the conclusions? The manuscript must describe methodologically and ethically rigorous research with conclusions that are appropriately drawn based on the data presented.

Reviewer #1: Partly

Reviewer #2: Partly

2. Has the statistical analysis been performed appropriately and rigorously?

Reviewer #1: I don't know

Reviewer #2: No

3. Have the authors made all data underlying the findings in their manuscript fully available (please refer to the Data Availability Statement at the start of the manuscript PDF file)?

Reviewer #1: Yes

Reviewer #2: No

4. Is the manuscript presented in an intelligible fashion and written in standard English?

Reviewer #1: Yes

Reviewer #2: Yes

5. Review Comments to the Author

Reviewer #1: Introduction

Shouldn’t “first 95” be defined more clearly p. 3 line 30: “they continue to lag behind other populations in the first 95 in Eastern….”?

Methods

What are “emancipated minors (16-<18 years)”?

p. 5 l. 69 “peer-to-peer model for KPs” – doesn’t this introduce a bias? From social network studies it looks like information spreads to people who like you and are part of the same mindsets?

p. 5 l. 72 “to their partners with unknown HIV status at home…” – how do they know it is unknown? Could this be a cause for overestimation of effect? -Is there a possible bias in the fact that knowing who the father (?) or partner is/having a partner could be a social determinant for health, empowerment or health literacy?

p.5 l. 81. How was partner violence handled in terms of home testing? – is it likely that a violent partner would accept home testing by partner?

p. 5 l. 84 “KP peers trained in HIVST, distributed HIVST kits in the community to KPs” – what does this mean exactly? How were KPs with unknown HIV status identified? How were FSW and MSM identified and how likely are they to openly identify as such in Uganda? Were there no fears of being registered for other purposes?

p. 5 l. 87 “KP peers trained in HIVST, distributed HIVST kits in the community to KPs” – could there be other ‘convenient meeting points for KPs?

Not clear to me if partners are also included in MSM/FSW ?

Results

p. 6 l. 109 These figures are not clear to me: “Most (6965 [74%]) to women aged 110 16 to 24 years (4496 [48%])” – what is 6965 and what is 4496?) . it is clearer in the table so the text is too condensed.

p. 6 l. 110 “Kits used by male sexual partners of mothers were 5212 (56%)” – but % of which number?

Discussion

What is meant by “early observations”? – please clarify

Would partners/MSM/FSW that know (or suspect) to be infected be more likely not to participate?

Some of the shortcomings in methods and results are caused by the fact that the study is based on routine data that were collected for general purposes. What would have been done differently if the study had been designed to collect the relevant data?

Reviewer #2: Nasuuna et al

This paper provides findings from an HIV self-testing (HIVST) programme focussed on key populations and sexual partners of pregnant mothers in Kampala, Uganda, reporting HIV prevalence by group and subsequent rates of linkage to HIV treatment for those testing HIV positive. This is an important research area, and the paper is generally well written. However I do have a number of points which hopefully would improve the analysis and its presentation, particularly regarding full presentation of all the relevant data.

Major comments

Throughout the manuscript, percentage linked to care is presented using the denominator of PLHIV confirmed HIV positive. I think authors should present percentage linked to care using the denominator of all those testing positive using HIVST, so there should be two linkage percentages reported for each group: percentage with HIVST-positive results whose HIV positivity is confirmed and percentage of those linked to care. Only half (126/252) of HIVST-positive individuals linked to care. This deserves more prominence. The Discussion (p9, line 155) states that linkage is higher than reported in Malawi (56%, ref 23) and South Africa (67%, ref 14). I have quickly checked ref 23, and the denominator for this seems to be all those testing positive by HIVST – so actually higher than the linkage reported here. I have not checked ref 14, but I strongly advise the authors to do so and check that they are comparing like with like.

The reason for the drop from 252 HIV-positives by HIVST to 170 confirmed HIV-positive should be made much clearer – how many of the 252 returned for confirmatory testing and how many were treated as HIVST false positives? The diagnostic accuracy of HIVST is also not stated (sensitivity, specificity) and its implications not discussed.

What percentage didn’t bring back the OraQuick paddle? What percentage didn’t present for confirmatory testing and were tested through a community visit?

Methods p5 line 70: how were the peers recruited?

Only those of unknown HIV status were offered HIVST. This deserves discussion. HIV status is dynamic, and it may be known to a mother’s sexual partner, but not necessarily to the mother. 17 of those reporting their HIV status already knew they were HIV positive but had not disclosed it to their partners/KP peers – surely even more individuals who report being HIV negative are actually not disclosing their status, or are unaware of their status?

Methods p5 line 88: What number of KPs refused to provide a telephone number, and if so, were they just excluded from the analysis or were they also refused a kit?

Results: I would recommend a flow chart figure to summarise the numbers of peer and mother distributors, numbers of kits distributed and number of recipients, stratified by KP type (FSW, MSM) and sexual partner. The flowchart could then show numbers and % with HIVST-positive results, numbers/% for confirmatory testing and finally, numbers/% starting HIV treatment. It would make the outcomes much clearer. The table could also be made more informative by adding extra columns – the first for all, then further columns for FSW, MSM and sexual partners of mothers.

It seems that 100% of the 9378 HIVST kits are accounted for, with recipients reporting their results. Surely there were some kits distributed and lost. Please report these in the text and/or flowchart.

I think the Discussion needs to go into more detail on why people are lost at each step in the pathway and the limitations of the analysis (e.g. only including those with unknown HIV status). Once there is more transparent reporting on numbers excluded, number of kits distributed per KP distributor etc, the Discussion should then make some fuller explanations for why linkage is so poor and suggest next steps to try to improve it. I think using large parts of the Discussion to compare HIV positivity rates with other countries’ (p9, first paragraph) is of less utility.

The figure could be presented in a more compelling way. In its current form, the 79% and 99 figures overlap and it’s unclear (changing the scale of the right hand y axis would solve that). However I think it would look better as a stacked bar chart, one bar for each population (FSW, MSM and partners) and either all the same height (so y axis is percentages, and numbers of PLHIV could be written on the chart), or different heights (y axis is number of PLHIV and percentages are written on the chart).

Minor comments

Introduction line 3: “in the first 95” sounds a little unclear here, out of context – it’s only explained later, on p33.

Last paragraph of the Introduction: authors cite refs 18 and 19 reporting HIVST for KP and mothers’ partners in Uganda. They should use this point in the paper to explain what insights their analysis provides above and beyond these two papers.

Methods p5 line 82: mothers at risk of IPV were “supported accordingly”. Please give brief details of this support and state whether these mothers were then excluded.

Methods p5 line 86: What are KP dens, and would these be different for FSW and MSM populations?

Methods p6 line 91: “mothers and peers were asked to confirm delivery” – how? Telephone?

Methods p6 line 94: Probably could delete “immediately”, since later in the sentence the time window is reported as “within two days”.

Discussion, 2nd sentence: “facility-based distribution was more utilized…” I think we need to see more data to confirm this. Firstly, make it clearer in the table that facility-based distribution was for mothers, and community-based distribution was for KPs. I would also want to know how many mothers and KPs were offered but declined tests (and what proportion were deemed ineligible for having no partners/peers, or only partners/peers of known HIV status). Also how many distributors were there in each group?

Discussion p9 line 148: Authors state that Malawi has a higher HIV prevalence than Uganda at 10.6%. State the prevalence in Uganda here as well. The discussions in this paragraph of positivity rates: worth quoting the uncertainty bounds.

Discussion: I’d recommend cost-effectiveness analysis as a future research direction.

Discussion last sentence: Rephrase “policy for improvement”.

6. PLOS authors have the option to publish the peer review history of their article (what does this mean?). If published, this will include your full peer review and any attached files.

**Do you want your identity to be public for this peer review?** For information about this choice, including consent withdrawal, please see our Privacy Policy.

Reviewer #1: **Yes: **Morten Sodemann

Reviewer #2: No

---

## [Editor Report · Decision Letter 1]

16 Nov 2021

PGPH-D-21-00581R1

Early Observations from the HIV Self-Testing Program among Key Populations and Sexual Partners of Pregnant Mothers in Kampala, Uganda: A Cross Sectional study

Dear Dr. Nasuuna,

Thank you for submitting your manuscript to PLOS Global Public Health. After careful consideration, we feel that it has merit but does not fully meet PLOS Global Public Health’s publication criteria as it currently stands. Therefore, we invite you to submit a revised version of the manuscript that addresses the points raised during the review process.

We look forward to receiving your revised manuscript.

Kind regards,

Christian Wejse, MD, PhD, Assoc.Prof

Academic Editor

Journal Requirements:

Additional Editor Comments (if provided):

The STROBE checklist only states that the items are completed, but it should be added where in the manuscript, so it is possible to evaluate if the item is properly dealt with
---

## [Editor Report · Decision Letter 2]

22 Nov 2021

Early Observations from the HIV Self-Testing Program among Key Populations and Sexual Partners of Pregnant Mothers in Kampala, Uganda: A Cross Sectional study

PGPH-D-21-00581R2

Dear Dr. Nasuuna,

We're pleased to inform you that your manuscript has been judged scientifically suitable for publication and will be formally accepted for publication once it meets all outstanding technical requirements.

Within one week, you'll receive an e-mail detailing the required amendments. When these have been addressed, you'll receive a formal acceptance letter and your manuscript will be scheduled for publication.

An invoice for payment will follow shortly after the formal acceptance. To ensure an efficient process, please log into Editorial Manager at https://www.editorialmanager.com/pgph/ click the 'Update My Information' link at the top of the page, and double check that your user information is up-to-date. If you have any billing related questions, please contact our Author Billing department directly at authorbilling@plos.org.

Kind regards,

Christian Wejse, MD, PhD, Assoc.Prof

Academic Editor